# Successful Management of and Recovery from Multiple Cranial Nerve Palsies following Surgical Ventral Stabilization in a Dog with Atlantoaxial Subluxation

**DOI:** 10.3390/vetsci9070322

**Published:** 2022-06-27

**Authors:** Joong-Hyun Song, Tae-Sung Hwang, Dong-In Jung, Hee-Jun Jeong, Chan Huh

**Affiliations:** 1Department of Veterinary Internal Medicine, College of Veterinary Medicine, Chungnam National University, Daejeon 34134, Korea; 2Institute of Animal Medicine, College of Veterinary Medicine, Gyeongsang National University, Jinju 52828, Korea; hwangts@gnu.ac.kr (T.-S.H.); jungdi@gnu.ac.kr (D.-I.J.); 3Ulsan S Animal Medical Center, Ulsan 44726, Korea; 24smart@naver.com (H.-J.J.); 24sacc@naver.com (C.H.)

**Keywords:** atlantoaxial subluxation, dog, polymethylmethacrylate, tongue paralysis, ventral stabilization

## Abstract

**Simple Summary:**

Ventral-approach techniques have been broadly employed for the surgical stabilization of atlantoaxial subluxation. The postoperative complications following ventral stabilization may include upper-respiratory, pharyngeal, and laryngeal dysfunction due to inadvertent damage to adjacent structures, such as the larynx, trachea, and neuronal tissues. The exact causes and management methods for postoperative complications after ventral stabilization have not yet been fully elucidated. Implanted polymethylmethacrylate cement can cause multiple cranial nerve palsies affecting the adjacent nuclei of the cranial nerves or their peripheral roots or axons. Early revision surgery may promote the full recovery of the affected neurological dysfunctions. Clinicians should consider cranial nerve palsies as major complications of ventral stabilization surgery in patients with atlantoaxial subluxation and be cautious about the volume and extent of polymethylmethacrylate during ventral stabilization, especially in very-small-breed dogs.

**Abstract:**

A 4-year-old spayed female miniature poodle dog presented with a 1-week history of acute tetraparesis. A neurological examination revealed severe neck pain and non-ambulatory tetraparesis. Computed tomography and magnetic resonance imaging showed hypoplastic dens with moderate compression of the spinal cord at C1–C2. The atlantoaxial subluxation (AAS) was surgically stabilized using ventral pins and polymethylmethacrylate (PMMA) cement. On the second postoperative day, the patient showed significant dyspnea, and aspiration pneumonia was identified on radiography. The patient exhibited dysphagia with abnormal food prehension and an inability to protrude the tongue, with no gag reflex. We tentatively diagnosed the patient with multiple cranial nerve (CN) palsies involving the 9th, 10th, and 12th CNs following surgical ventral stabilization. The protruding cranial part of the implanted PMMA cement, which could mechanically contribute to the corresponding CNs dysfunction, was surgically removed. The symptoms gradually improved, and the patient showed normal tongue movement 1 month after revision surgery. In conclusion, we report herein a canine case of multiple CN palsies following ventral stabilization surgery for AAS. The experience gained from this case suggests an optimized management plan for postoperative neurological complications associated with ventral stabilization.

## 1. Introduction

Atlantoaxial subluxation (AAS) is a widely recognized developmental cervical myelopathy that primarily affects young toy-breed dogs. Congenital AAS typically results from a loss of atlantoaxial (AA) ligamentous support, often with concurrent agenesis, hypoplasia, or dens malformation [1]. Acquired traumatic subluxations may occur in any breed of dog or cat, usually due to a rupture of the AA ligaments or a fracture of the dens [2]. Patients’ clinical signs usually range from mild neck pain to paralysis, depending on the severity of the spinal cord compression from subluxation [3]. Regardless of the pathogenesis, surgical stabilization is the preferred treatment option for dogs with prominent signs of neurological dysfunction due to AAS.

Surgical treatment for AAS includes both dorsal and ventral stabilization techniques, and ventral stabilization is most often preferred because of the better visualization and accessibility of the AA junction and lower complication rate [4]. However, despite the relative safety of ventral stabilization, complication rates after surgery have been documented to be as high as 33–53% [5]. The postoperative complications following ventral stabilization may include upper respiratory, pharyngeal, and laryngeal dysfunction (e.g., coughing, gagging, laryngeal paralysis, dyspnea, and aspiration pneumonia) due to inadvertent damage to adjacent structures, such as the larynx, trachea, and neuronal tissues [6,7]. Only some patients with mild respiratory complaints (coughing and dyspnea) recover spontaneously, and those suspected of having intraoperative caudal brainstem trauma have a poor prognosis. Nevertheless, the exact causes and management methods for postoperative complications after ventral stabilization have not yet been fully elucidated.

We report here a canine case of multiple cranial nerve (CN) palsies following surgical ventral stabilization for AAS. Although dogs with postoperative complications following ventral stabilization have been described in previous reports [6,7], to the best of our knowledge, CN palsies with tongue paralysis have not yet been reported in the veterinary literature. Clinicians should consider CN palsies as major complications of ventral stabilization for AAS.

## 2. Case Presentation

A 4-year-old spayed female miniature poodle dog weighing 3.1 kg presented with a 1-week history of acute tetraparesis. The dog was generally healthy until the clinical signs were observed. She had been fully vaccinated and dewormed, with no history of trauma or toxin exposure. The neurological examination revealed severe neck pain and tetraparesis with signs of upper motor neurons in the thoracic and pelvic limbs. Except for these neurological abnormalities, no other abnormalities were noted on the physical examination. The results of the complete blood-cell count (ProCyte Dx, IDEXX Laboratories, ME, USA) and serum biochemistry (Catalyst One Chemistry Analyzer, IDEXX Laboratories, ME, USA) were unremarkable. On the lateral flexed radiographs, the space between the spinous process of the axis and the dorsal arch of the atlas was increased (Figure 1A).

The dog underwent computed tomography (CT) and magnetic resonance imaging (MRI) to evaluate for suspected cervical myelopathy. The CT (Action 16-slice scanner, Toshiba Medical Systems, Otawara, Japan) revealed hypoplastic dens of the axis with prominent compression of the spinal cord at the levels of the first and second cervical vertebrae (Figure 1B). The MRI (Vantage Atlas 1.5-T scanner, Toshiba Medical Systems, Otawara, Japan) at the same anatomical level as the CT scan revealed that the spinal cord was moderately compressed, which in turn caused inflammatory changes in the adjacent spinal-cord parenchyma (Figure 1C). Based on these findings, congenital AAS with hypoplasia of the dens was diagnosed.

A surgical stabilization was performed via ventral fixation using cortical screws and polymethylmethacrylate (PMMA). To establish safer surgical stabilization, we utilized a 3D printing drill-guide template (CUSTOMEDI, Daejeon, Korea). AA trans-articular fixation was achieved with cortical screws (1.5 mm in diameter and 8 mm or 12 mm in length) under the guidance of a 3D-printed template. PMMA (Spinofill, Injecta, Gunpo, Korea) was prepared according to the manufacturer’s instructions and placed on the ventral aspect of the AA joint to enclose the entire screw. Postoperative radiographs revealed adequate alignment and fixation.

The dog’s neck pain declined rapidly 1 day after surgical stabilization, and the patient was able to transiently bear weight without medical or physical assistance. On the second postoperative day, the patient developed significant dyspnea, cough, hypersalivation, and hyperthermia (Figure 2A). The patient’s C-reactive protein (CRP) level increased to 3.6 mg/L (reference range, 0.1–1.0 mg/L). The radiographs revealed severe alveolar infiltrate in the right middle lobe, consistent with acute aspiratory pneumonia. Furthermore, the patient exhibited dysphagia, with abnormal swallowing of food and an inability to protrude the tongue. The patient continuously wanted to eat but was unable to prehend and masticate food by herself. Moreover, difficulty in swallowing saliva and the accumulation of airway secretions worsened the dog’s breathing. C-arm fluoroscopy (OSCAR Prime, GENORAY, Sungnam, Korea) revealed that the patient was unable to swallow semi-solid food or water. A detailed CN assessment showed that the pharyngeal reflex was prominently decreased, with no swallowing or gag reflex, which was evaluated by applying pressure to the neck and tongue. The tongue did not voluntarily move and retract after the tongue had been manually stretched. No tongue atrophy or deviation were observed. Accordingly, we tentatively diagnosed the patient with multiple CN palsies involving the 9th, 10th, and 12th CNs following surgical ventral stabilization.

A tracheostomy tube (Yunshin Medical Co., Bucheon, Korea) and an esophageal feeding tube (Sewoon Medical Co., Cheonan, Korea) were placed to alleviate the clinical signs of dyspnea and to assist enteral feeding, respectively. A constant-rate infusion of cefotaxime (Kukje Pharm, Sungnam, Korea) was initiated at 2 mg/kg/h following a loading dose of 20 mg/kg. Butorphanol was administered intravenously at a dose of 0.2 mg/kg every 6 h. Oral prednisolone was also initiated at 0.25 mg/kg twice a day to improve surgery-induced neuroinflammation. Oropharyngeal secretions and tracheal sputum were aspirated through a tracheostomy tube every 6 h. A soft canned diet (Recovery, Royal Canin Veterinary Diet, MO, USA) was given four times a day for 15 min through the esophageal feeding tube. The oral cavity was then rinsed twice daily with 0.12% chlorhexidine gluconate (Hexamedin; Bukwang Pharm., Seoul, Korea). On the 6th postoperative day, the patient achieved full weight bearing and became mobile without any signs of pain. On the repeated radiographs on the same day, the patient was shown to have recovered from aspiratory pneumonia. However, the multiple CN palsies did not show any significant improvement, and the tongue gradually protruded to the right lateral side (Figure 2B). The multiple CN palsies thought to involve the 9th, 10th, and 12th CNs did not improve and slowly deteriorated even at 2 weeks postoperatively.

Considering the anatomical features of these three CNs, the neurological abnormalities in the patient were strongly suspected to be postoperative complications associated with the implanted PMMA cement for ventral stabilization. This suspicion was supported by the radiographic results (Figure 3A,B). Because of the proximity of the cranial part of the implanted PMMA to the ventral brainstem centers and projections of the multiple CNs, it was judged that the PMMA cement protruding beyond the level of the cranial aspect of the atlas would obviously compress the corresponding neuronal structures of the CNs.

Thus, we decided to surgically remove the protruding cranial part of the implanted PMMA cement located just below the tympanic bulla. Surgical access to the ventral aspect of the cervical spine was performed, and the protruding cranial part of the implanted PMMA cement was removed approximately 7 mm from the level of the cranial aspect of the atlas (Figure 3C). After the revision surgery, suction of the oropharyngeal and tracheal secretions and oral rinsing with 0.12% chlorhexidine gluconate were continued. Furthermore, repeated mechanical stimulation of the entire tongue using a tongue pressor and food was performed three times a day for the rehabilitation and evaluation of the tongue’s motility. Five days after the revision surgery, there was no improvement in the tongue’s movement. Twelve days after the revision surgery, the tongue moved slightly with a mild gag reflex after the application of pressure, and the protruded tongue returned to its normal position. Twenty days after the revision surgery, there was a significant improvement in the gag reflex, allowing the tip of the tongue to reach the lip area (Figure 2C). Placing a small amount of food in the caudal part of the patient’s mouth made it possible for the patient to prehend and swallow food by herself. One month after the revision surgery, the tongue movement and swallowing function returned to normal, and the neurological signs of AAS, including neck pain and tetraparesis, completely disappeared. The tracheostomy and esophageal feeding tubes were removed, and the patient was successfully discharged.

## 3. Discussion

Ventral approach techniques have been broadly employed for the surgical stabilization of AAS. Central-nervous-system injury is a well-documented complication associated with ventral stabilization; it often results in upper respiratory and cardiac problems, including dyspnea, laryngeal dysfunction, and cardiac arrest. These complications may occur within the first 48 h postoperatively due to perioperative neuronal trauma by surgical manipulation and can lead to overt clinical dysfunctions, ranging from transient reversible respiratory complaints to instant death [7]. It has been suggested that the secondary mechanical damage caused by implanted PMMA cement, as well as the damage caused by surgical manipulation during ventral stabilization, may cause serious complications in very small dogs [6]. A relatively high volume of implanted PMMA cement is required in small dogs, which easily leads to space-occupying damage to the caudal brainstem center. However, there is no compelling evidence to determine the prevalence of implanted PMMA-induced complications or to develop an optimal management plan after complications.

We first expected that the neurological signs of this patient would be transient damage to the neuronal structures by surgical manipulation; however, the symptoms deteriorated over time. Thus, we decided to revise the surgery in consideration of the permanent neuronal damage caused by the excess volume of the implanted PMMA cement. The neurological symptoms were considered consistent with a lesion in the caudal brainstem and the possible multiple involvement of the nuclei of the 9th, 10th, and 12th CNs or their peripheral roots or axons. The neuronal cell bodies of CNs 9 (glossopharyngeal nerve) and 10 (vagus nerve) are located in the medulla oblongata of the brainstem and leave the cranial cavity through the jugular foramen and tympano-occipital fissure [8]. The neuronal cell bodies of CN 12 (hypoglossal nerve) are also located in the medulla oblongata of the brainstem and leave the cranial cavity through the hypoglossal canal [8]. The neurons of CN 9 provide pharyngeal branches and innervate the styloglossus muscle and other pharyngeal muscles for motor and sensory functions, and they also provide parasympathetic fibers for salivary and taste functions.

The neurons of CN 10 provide pharyngeal and laryngeal branches and innervate the palate, pharynx, cervical esophagus, and larynx for motor and sensory functions. They also provide parasympathetic innervation to the gastrointestinal tract and some fibers for taste and salivation. The neurons of CN 12 course to innervate the tongue muscles and geniohyoideus for the motor function of the intrinsic muscles of the tongue. Lesions in CNs 9 and 10 result in the impairment of laryngeal function and varying degrees of difficulty in swallowing and gagging [9]. Lesions in CN 12 result in the impairment of tongue function in prehension, deglutition, mastication, and vocalization [9]. Considering the anatomical location of these CNs and the neurological dysfunction of the patient discussed in this study, multiple CN palsies involving the 9th, 10th, and 12th CNs due to the implanted PMMA cement were strongly suspected. A definitive diagnosis was made by confirming the recovery from the multiple CN palsies after revision surgery. Based on our experience with this patient, more caution is needed in the use of PMMA cement in ventral stabilization for small canine patients with AAS, and the extent of implanted PMMA cement should be measured, as it may cause serious neurological problems. Moreover, in patients with multiple CN palsies after PMMA cement implantation, early revision surgery may promote recovery from neurological dysfunction.

CN dysfunction due to mechanical injury has rarely been reported in veterinary medicine. A previous case report of a dog with traumatic atlanto-occipital luxation (5 days from the trauma event to the initial presentation) documented that the neurological dysfunction of the lingual paresis fully recovered 1 day after the closed reduction of the luxated articulation [10]. An experimental study of cochlear nerve injuries induced by mechanical manipulation for a short period showed the transient loss and rapid recovery of cochlear nerve function [11]. Regarding CN dysfunction due to a reversible pathological etiology, if there is no cause of additional nerve damage in canine patients with CN dysfunction, neurological symptoms generally resolve within a month with specific treatment or spontaneous recovery [12,13]. In the present case, mechanical damage to the affected CNs lasted approximately 2 weeks, and it took approximately 1 month from the revision surgery for the patient to fully recover from the multiple CN palsies. Therefore, we surmised that the recovery from CN damage can be reversed by resolving the underlying etiologies, and the complete recovery of the functions of the affected CNs may take approximately a month.

The application of trans-articular ventral pins and PMMA cement for the management of AA instability is a widely recommended surgical method for providing greater stability in AA fixation [5,14]. Although PMMA implantation is useful, some studies have revealed the disadvantages of its use in AASs, including thermal damage, the risk of infection, necrosis of the adjacent structures, and the hindrance of neck movement, and have argued for the need to minimize the volume and size of PMMA [15]. However, there are no reported data on the appropriate amount or extent of PMMA cement for ventral stabilization according to the size of the patient. Thus, further investigation is needed to establish a guideline for the proper use of PMMA cement in ventral stabilization surgery for patients with AAS.

## 4. Conclusions

This report describes a canine case of multiple CN palsies following ventral stabilization surgery for AAS. The implantation of PMMA cement can cause multiple CN palsies affecting the adjacent nuclei of the CNs or their peripheral roots or axons. Early revision surgery may promote the full recovery of the affected neurological dysfunctions. Clinicians should consider CN palsies as major complications of ventral stabilization surgery in patients with AAS and be cautious about the volume and extent of PMMA during ventral stabilization, especially in very-small-breed dogs.

## Figures and Tables

**Figure 1 vetsci-09-00322-f001:**
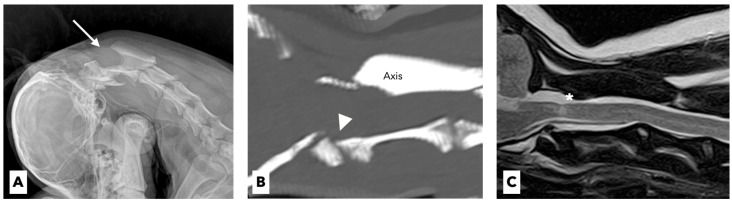
Lateral flexed radiographic image (**A**) showing that the space (arrow) between the spinous process of axis and the dorsal arch of atlas is increased. Sagittal-plane CT imaging (**B**) reveals a hypoplastic dens (arrowhead) of the axis with prominent compression of the spinal cord. T2-weighted magnetic resonance image (**C**) reveals moderately compressed spinal cord and inflammatory changes of the adjacent spinal cord parenchyma (asterisk).

**Figure 2 vetsci-09-00322-f002:**
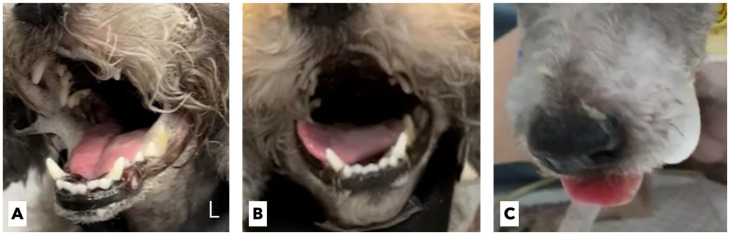
Gross images of the tongue over time. (**A**) The patient is not able to protrude or move its tongue, and hypersalivation is identified. (**B**) On the 6th postoperative day, the tongue is gradually protruding to the right lateral side. (**C**) Twenty days after revision surgery, there is significant improvement in the tongue movement, with the tip of the tongue reaching the lip area.

**Figure 3 vetsci-09-00322-f003:**
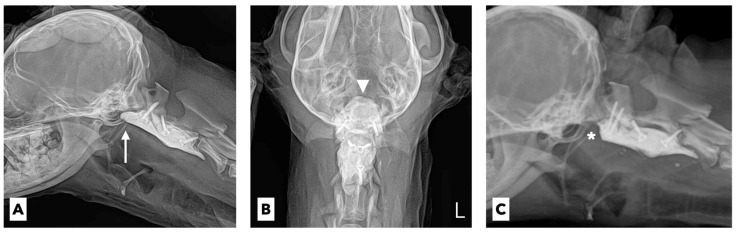
Radiographic images before (**A**,**B**) and after (**B**) revision surgery. (**A**,**B**) The PMMA cement protrudes beyond the level of cranial aspect of the atlas (arrow and arrowhead). (**C**) The protruded cranial part of the implanted PMMA cement was successfully removed through revision surgery (asterisk).

## Data Availability

Not applicable.

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
