# Peer review of "Successful Management of and Recovery from Multiple Cranial Nerve Palsies following Surgical Ventral Stabilization in a Dog with Atlantoaxial Subluxation"

_vetsci, 2022, doi:10.3390/vetsci9070322_

Round 1
Reviewer 1 Report
This case report accurately describes a major complication following ventral stabilisation using a plate and screws and PMMA. In the literature, as the author correctly mentions, there are no references to the amount of PMMA to be used, but in reference surgery texts it is stated that as for all ventral techniques
the minimum amount of polymethylmethacrylate necessary to cover all the screws should be used, because it can cause significant compression of surrounding soft tissues (VETERINARY SURGERY: SMALL ANIMAL, SECOND EDITION ISBN: 978-0-323-32065-8 Vol 1: 978-0-323-50966-4, chapter 31).
I would therefore suggest that the authors present this paper more as a major complication encountered than as a demonstration that PMMA-associated brainstem compression involving multiple cranial nerve nuclei can resolve within 1 month, and that the experience gained rather serves to understand how preoperative and intraoperative evaluation are crucial for successful outcome.
In addition, I also suggest these further observations:
Line 13: tetraparesis ambulatory or nonambulatory?
Line 55: “previous report” I believe that the bibliography consulted in support of the statement should be cited at this point
Line 67: “however” seems to put in direct relation the bloodwork results and the radiographic alterations found. Besides, the radiographic image in figure 1 seems to be in flexed position, the authors should describe better the radiographic positioning which they applied and review the caption of figure 1.
Line 96: could you say what you mean by rapidly? (hours, days?). Was it on pain-killers?
Line 131-132 Reading this sentence, the question arises: so the dog, after the first post-surgery day in which you claimed (line 97) he was able to bear weight without assistance, deteriorated again with the truncus encephalic signs and only on the sixth day was he able to recover again?
Line 132: when did she recover from aspiration pneumonia? Did you repeat a xray scan of the chest?
Line 138: Author should write that the dog underwent xray scan to assess the problem.
Line 143: “nuclei” the ambiguous nucleus, where the somatic motor neurons of the cranial nerves IX, X and XI are contained, is placed in the intracranial area while the PMMA appears, from the attached images, in the extracranial area; on the basis of what observation do the authors hypothesize the compression of this nucleus?
Line 244: I think it's risky to generalise about the recovery time: in your case it took a month. I recommend rephrasing the sentence, as, by the way, you only have a clinical case so it's not possible to draw conclusions.
Figure 1a: Is it assumed that the x-ray was taken in narcosis? it's correct? If so, it seems that the patient does not have an orotracheal tube: the flexion maneuver of the neck, during atlantoaxial instability, exposes the dog to risk of respiratory arrest and sudden death. Furthermore, the unprotected operator's hand is against all radiation protection rules (same problem in figures 3a and 3c). Both aspects make the image unsuitable for publication for both scientific and educational reasons: do the authors have no alternative radiographic images to be published?
Check the bibliography, some authors' names are not reported correctly
Author Response
Dear editor
We thank you for your time and consideration on our submission. Thus, it is with great pleasure that we resubmit our article for further consideration. We have incorporated changes that reflect the detailed suggestions you have graciously provided. Revisions made after carefully considering the comments of the reviewers’ and editor are as follows. The appropriate changes made in the revised manuscript are highlighted. We believe that these modifications have strengthened the manuscript and hope that the revised manuscript is suitable for publication in the Veterinary Sciences.
[Reviewer 1.]
This case report accurately describes a major complication following ventral stabilisation using a plate and screws and PMMA. In the literature, as the author correctly mentions, there are no references to the amount of PMMA to be used, but in reference surgery texts it is stated that as for all ventral techniques
the minimum amount of polymethylmethacrylate necessary to cover all the screws should be used, because it can cause significant compression of surrounding soft tissues (VETERINARY SURGERY: SMALL ANIMAL, SECOND EDITION ISBN: 978-0-323-32065-8 Vol 1: 978-0-323-50966-4, chapter 31).
I would therefore suggest that the authors present this paper more as a major complication encountered than as a demonstration that PMMA-associated brainstem compression involving multiple cranial nerve nuclei can resolve within 1 month, and that the experience gained rather serves to understand how preoperative and intraoperative evaluation are crucial for successful outcome.
Response: Thank you for your suggestion. We have revised the line 56-57 and 247-248 in line with your comment.
In addition, I also suggest these further observations:
Line 13: tetraparesis ambulatory or nonambulatory?
Response: Thank you for pointing this out. We have accordingly corrected this point (line 13).
Line 55: “previous report” I believe that the bibliography consulted in support of the statement should be cited at this point
Response: Thank you for pointing this out. We have accordingly corrected this point (line 55).
Line 67: “however” seems to put in direct relation the bloodwork results and the radiographic alterations found. Besides, the radiographic image in figure 1 seems to be in flexed position, the authors should describe better the radiographic positioning which they applied and review the caption of figure 1.
Response: Thank you for pointing this out. We have accordingly corrected this point (line 68-72).
Line 96: could you say what you mean by rapidly? (hours, days?). Was it on pain-killers?
Response: Thank you for pointing this out. We have accordingly corrected this point (line 99-100).
Line 131-132 Reading this sentence, the question arises: so the dog, after the first post-surgery day in which you claimed (line 97) he was able to bear weight without assistance, deteriorated again with the truncus encephalic signs and only on the sixth day was he able to recover again?
Response: Thank you for pointing this out. On day 1, the patient was able to “transiently” bear weight without medical and physical assistance. We have accordingly corrected this point (line 99-100).
Line 132: when did she recover from aspiration pneumonia? Did you repeat a xray scan of the chest?
Response: Thank you for pointing this out. We have accordingly corrected this point (line 135-136).
Line 138: Author should write that the dog underwent xray scan to assess the problem.
Response: Thank you for your suggestion. We have revised the line 144-145 in line with your comment.
Line 143: “nuclei” the ambiguous nucleus, where the somatic motor neurons of the cranial nerves IX, X and XI are contained, is placed in the intracranial area while the PMMA appears, from the attached images, in the extracranial area; on the basis of what observation do the authors hypothesize the compression of this nucleus?
Response: You have raised an important point and we agree with your suggestion. We have revised the line 145-148 in line with your comment.
Line 244: I think it's risky to generalise about the recovery time: in your case it took a month. I recommend rephrasing the sentence, as, by the way, you only have a clinical case so it's not possible to draw conclusions.
Response: Thank you for pointing this out. We have accordingly revised the line 244.
Figure 1a: Is it assumed that the x-ray was taken in narcosis? it's correct? If so, it seems that the patient does not have an orotracheal tube: the flexion maneuver of the neck, during atlantoaxial instability, exposes the dog to risk of respiratory arrest and sudden death. Furthermore, the unprotected operator's hand is against all radiation protection rules (same problem in figures 3a and 3c). Both aspects make the image unsuitable for publication for both scientific and educational reasons: do the authors have no alternative radiographic images to be published?
Response: You have raised an important point and we agree with your suggestion. We apologize for the unsuitable images. Unfortunately, we don’t have alternative image of 1A. We have attempted to do our best for revising the Fig 1A. and Fig 3 (Line 71 and 151).
Check the bibliography, some authors' names are not reported correctly
Response: Thank you for pointing this out. We have accordingly corrected this point (line 277-305).

Reviewer 2 Report
This manuscript describes a canine case of multiple CN palsies following ventral stabilization surgery for atlantoaxial subluxation.
After carefully reading the manuscript, I conclude that the abstract, introduction, and other chapters cover the issues discussed in an extensive and proper manner. The conclusions presented by the authors are consistent with the evidence and relate to the main research issue.
I recommend publication accept in present form.
Author Response
Dear editor
We thank you for your time and consideration on our submission. Thus, it is with great pleasure that we resubmit our article for further consideration. We have incorporated changes that reflect the detailed suggestions you have graciously provided. Revisions made after carefully considering the comments of the reviewers’ and editor are as follows. The appropriate changes made in the revised manuscript are highlighted. We believe that these modifications have strengthened the manuscript and hope that the revised manuscript is suitable for publication in the Veterinary Sciences.
[Reviewer 2.]
This manuscript describes a canine case of multiple CN palsies following ventral stabilization surgery for atlantoaxial subluxation.
After carefully reading the manuscript, I conclude that the abstract, introduction, and other chapters cover the issues discussed in an extensive and proper manner. The conclusions presented by the authors are consistent with the evidence and relate to the main research issue.
I recommend publication accept in present form.
Response: Thank you for your reviewer report.

Reviewer 3 Report
Thank you for the opportunity to review this article. The report describes a case of surgical ventral atlanto-axial stabilisation using pins and PMMA. I think the report is extremely interesting. In particular, the diagnosis carried out by correlating the symptoms attributable to cranial nerve suffering and the position of the PMMA. From a technical point of view: the introduction provides the necessary elements required to understand the issue; the methodology is clear; the case well presented; the discussion interesting and detailed; the conclusions consistent. I think the report adds useful elements to improve the use of PMMA in ventral atlanto-axial stabilisation, particularly with regard to the position and size of the PMMA, of which the literature does not provide clear indications. I therefore consider the article worthy of publication. I would only have one request: to include a short paragraph in the discussion regarding existing surgical options for the management of atlanto-axial subluxations, including relative better indications, pros and cons. Thank you.
Round 2
Reviewer 1 Report
Dear Authors,
important errors persist in the bibliography which must be corrected absolutely. I recommend checking all the bibliography with particular attention to the author's name and surname and to the number of editions of books.
Author Response
Thank you for pointing this out. We have accordingly revised the line 264-291.
